# Sedentary Behavior Counseling Received from Healthcare Professionals: An Exploratory Analysis in Adults at Primary Health Care in Brazil

**DOI:** 10.3390/ijerph19169963

**Published:** 2022-08-12

**Authors:** André Snége, Alexandre Augusto de Paula da Silva, Grégore Iven Mielke, Cassiano Ricardo Rech, Fernando Carlos Vinholes Siqueira, Ciro Romelio Rodriguez-Añez, Rogério César Fermino

**Affiliations:** 1Research Group on Environment, Physical Activity and Health, Federal University of Technology-Paraná, Curitiba 81310-900, Brazil; 2Postgraduate Program in Health Sciences, Research Group on Physical Activity and Quality of Life, Pontifical Catholic University of Paraná, Curitiba 80215-901, Brazil; 3School of Public Health, The University of Queensland, Brisbane, QLD 4006, Australia; 4Research and Study Group in Urban Environment and Health, Federal University of Santa Catarina, Florianopolis 88040-900, Brazil; 5Postgraduate Program in Physical Education, Federal University of Pelotas, Pelotas 96055-630, Brazil; 6Postgraduate Program in Physical Education, Federal University of Paraná, Curitiba 81531-980, Brazil

**Keywords:** sitting time, directive advice, epidemiological studies, primary health care, public health

## Abstract

Counseling by health professionals has promising results in behavior change and is recommended as part of integrated community interventions. However, the knowledge about sedentary behavior (SB) counseling is incipient. The study aimed to identify the prevalence and explore the associated factors with SB counseling received from healthcare professionals by adults in primary health care (PHC) in Brazil. A cross-sectional study was conducted in 2019 that included a representative sample of 779 users in all 15 basic health units (BHU) in São José dos Pinhais, Paraná. We identified those who reported having received SB counseling during a consultation. The association between the sociodemographic factors, chronic diseases, access to health services, physical activity, SB, and counseling were analyzed using Poisson regression in a hierarchical model. The prevalence of counseling was 12.2% (95% CI: 10.1–14.7%); it was higher in women (PR: 1.77; 95% CI: 1.10–2.83), those aged ≥60 yrs (PR: 1.84; 95% CI: 1.14–2.98), BMI ≥ 30 kg/m^2^ (PR: 2.60; 95% CI: 1.31–5.17), who consume ≥3 medications (PR: 2.21; 95% CI: 1.06–4.59), and those who spend a prolonged period of the day engaged in SB (4th quartile PR: 3.44; 95% CI: 1.88–6.31). The results highlight that SB counseling is underutilized and incipient in PHC. Understanding these results can help managers and healthcare professionals in BHU teams to implement and direct specific actions to reduce SB in adults through counseling.

## 1. Introduction

The changes in social and environmental factors and technological advances have contributed to an increase in time spent in sedentary behavior [SB; i.e., any waking behavior characterized by an energy expenditure ≤1.5 metabolic equivalents (METs), while in a sitting, reclining or lying posture] [1]. Global estimates indicate that adults engage in this behavior for between 6–8 h/day, possibly longer in high-income countries [2,3,4]. The burden of SB is estimated to have cost the UK around £8 billion, and about 69,276 deaths would have been avoided [5]. In Brazil, it is estimated that adults spend between 7–8 h/day in SB. It is associated with the male sex, younger age group, higher education and socioeconomic levels, and insufficient physical activity (PA) [6,7]. In combination with low PA, SB is associated with an increased risk of cardiovascular disease, cancer, and premature mortality [8,9,10]. 

The World Health Organization (WHO) suggested promoting PA within primary health care (PHC), and the counseling by health professionals has had promising results in behavior change and is recommended as part of integrated community interventions [11]. Moreover, several actions to prevent, manage and reduce SB have been recommended globally [12,13,14,15]. Environmental, behavioral, and multicomponent interventions have significantly reduced SB by 24–60 min/day [14]. Among these interventions, counseling by a healthcare professional is a promising strategy that can reduce SB by around 16–53 min/day [12,14,15]. Counseling is an educational activity that empowers people to manage their lifestyles [16,17,18]. The primary care policy has been implemented in the unified health system in Brazil, mainly in basic health units (BHU) [19]. Public health services are estimated to serve 70% of the population, with 47% of users of BHU and 76% of adults (≈160 million) consulting a physician recently [20]. Brazil has 42,488 BHUs, which could be suitable places for counseling, as the population has high confidence in guidance from healthcare professionals [16,19,20]. The BHUs are public clinics strategically distributed within a city, with free access to PHC provided by physicians, nurses, pharmacists, physiotherapists, nutritionists, psychologists, and community health agents [19]. In some Brazilian cities, some BHUs also have physical education professionals [21].

Only one study from an upper-income country has explored SB counseling in PHC by asking patients whether their primary care provider had inquired about their lifestyle behaviors about SB (e.g., sitting while watching TV or using the computer) [22]. The prevalence of counseling was 10%, and only obesity was associated with the outcome [22]. However, the specific context in an academic primary care clinic in Dallas (TX, USA) is not representative of the sociocultural reality of Latin American countries. It is essential to expand knowledge in BHU to identify those individuals most exposed to SB counseling [7,8,9,10,18]. This approach and the results are essential for healthcare professionals to implement and direct specific actions in groups that spend prolonged SB, especially in upper-middle-income countries [4,15,18,23]. 

Therefore, this study aimed to identify the prevalence and explore the associated factors with SB counseling received from healthcare professionals by users adults at PHC units in Brazil.

## 2. Materials and Methods 

### 2.1. Design, Study Site, and Ethical Aspects

Between April and October 2019, a quantitative, cross-sectional study was conducted on a representative sample of users adults who attended BHU in São José dos Pinhais [24,25]. This is a medium-sized developed city in Southern Brazil, with an estimated population of 329,000 inhabitants, located 18 km from the capital Curitiba. The city has an area of 946 km^2^ (79% rural), a populational density of 342 inhabitants/km^2^, and the Human Development Index is high (0.758) [25]. At the beginning of the planning of this study, the city had 27 BHU (56% in urban areas, n = 15) [24]. All BHU in urban areas were intentionally selected, as they correspond to access for 90% of the population. 

The study was approved by the Research Ethics Committee of the Pontifical Catholic University of Paraná (#2.882.260). Participants were consulted, informed about voluntariness, and agreed to participate in the research by signing an informed consent form.

### 2.2. Sample Size, Number of Participants, and Power

The number of participants was estimated based on the average number of visits to each BHU between January and February 2019 (N = 34,275) [25]. The sample calculation was conducted to represent the population considering a prevalence of PA counseling received through the PHC services of 30% (identified in the literature review), 95% confidence interval (CI); sampling error of four percentage points (4 p.p.), and design effect of 1.5 [26]. As a result, the minimum number of participants was estimated at 745. However, with a 10% increase in losses and refusals, we estimated the need to approach 820 participants. We decided to approach a surplus of 100 people (n = 920) to reduce the estimation errors for multivariate analyses in future studies. The sample size was proportionally calculated by the number of visits to each BHU and varied from 31 to 92 users.

As this study used secondary data from a major project, the number of participants required to represent the population was estimated a posteriori. Specifically, we considered the frequency of SB counseling obtained in a pilot study (17%), 95% CI, sampling error of 4 p.p., and design effect of 1.5. Therefore, the minimum sample required was 503 adults. 

### 2.3. Selection of Participants

The participants were selected based on their position in the BHU room while waiting for healthcare professionals’ consultation. People were counted from one to five, from left to right, starting at the BHU door. The third user was approached and invited to participate in the study [27], but in case of refusal or the participant did not meet the inclusion criteria, the first person on the left was selected. 

Only users adults (≥18 yrs old) were eligible and invited. Among them, we excluded people who lived outside the urban area (due to differences in lifestyle), were using the BHU for the first time, had some physical limitations for PA practice (e.g., wheelchair and crutch users), had some cognitive limitations that prevented comprehension of the questionnaire (e.g., hearing impairment, mental disorders) or phonation limitation from answering the interviews (n = 9). 

### 2.4. Data Collection

The face-to-face interviews were conducted by trained interviewers, before or after consultation with a health professional, who read the questionnaire in an individual and reserved room, ensuring no external influence on the answers [28]. The average time of interviews was 18 min (±5 min, 9–55 min).

### 2.5. Outcome Variable: Sedentary Behavior Counseling 

SB counseling was assessed based on the dichotomous answer (no, yes) to the question: “*During the last 12 months, once you were at a BHU, did you receive counseling to reduce your SB* (advice, tips, or guidance on sitting time to change/improve your health)?”. This measure was used in similar studies to assess PA counseling [22,25,27,29].

Before asking the question, the SB was explained to the participants as any activities performed in the sitting position, such as watching TV, working, using the bus or car for commuting, using the computer, or reading.

### 2.6. Independent Variables

The variables tested included sociodemographic characteristics, health conditions, access to health services, and health risk behaviors [1,2,7,9,22,30,31].

#### 2.6.1. Sociodemographic Characteristics

Age was classified into three groups (18–39, 40–59, and ≥60 yrs old). The marital status was assessed using three categories (single, married/stable union, divorced/widowed) and grouped into single (single, divorced, widowed) and not single (married, stable union). Skin color was self-reported into five categories (white, black, yellow, brown, and indigenous) and categorized into white and non-white (other categories) [32]. The socioeconomic status was assessed using a standardized questionnaire, and the participants were classified into low and high economic level [33].

#### 2.6.2. Health Conditions

Body Mass Index (BMI) was calculated (kg/m^2^) with self-reported body mass and height and classified as underweight or normal weight (≤24.9 kg/m^2^), overweight (25.0–29.9 kg/m^2^), or obese (≥30.0 kg/m^2^). The presence of chronic diseases was evaluated with a dichotomous answer established by the medical diagnosis self-report for eight diseases (hypertension, diabetes, dyslipidemia, coronary artery disease, circulatory diseases, osteoporosis, pulmonary disease, and depression) [34], which were added and classified into three categories (0, 1–2, ≥3). The participants reported the continuous medication prescribed for chronic diseases and were categorized into three groups (0, 1–2, ≥3) [25,27].

#### 2.6.3. Access to Health Services

Participants reported the frequency of visits to BHU in the past 12 months and were classified into three categories (0–2, 3–6, ≥7 times/year) [27].

#### 2.6.4. Health Risks Behaviors

##### Leisure-Time Physical Activity

The leisure-time PA was assessed by International Physical Activity Questionnaire [35]. Participants self-reported the weekly frequency and mean daily volume of walking, moderate and vigorous PA. The score in each activity/intensity was obtained in minutes per week (min/week) by multiplying the weekly frequency by the mean daily volume. The MVPA score was calculated using the minutes per week of moderate PA, added to the vigorous activities, multiplied by two [min/wk of moderate PA + (min/wk of vigorous PA*2)]. Walking and MVPA were operationalized into three categories (0–9, 10–149, ≥150 min/week) according to WHO recommendations [36].

##### Sedentary Behavior

The daily time spent engaging in SB was evaluated using a questionnaire developed and validated for the Brazilian context [7,37]. Participants reported the mean time spent sitting in daily activities in four domains (domestic, work, study, commuting) [1,7]. The mean score (h/day) for each activity was added to estimate the total sitting time [7,37], which was classified into quartiles to estimate the most exposed to SB (1st: 0–1.7; 2nd: 1.8–3.3; 3rd: 3.4–6.1; 4th: ≥6.2 h).

### 2.7. Data Quality Control

Quality control was assured in six steps. First, the interviewers (members of the research group, undergraduate and graduate students in physical education) received 20-h theoretical and practical training on the technical procedures of conducting the interviews [participant approach, record of losses (they agreed to participate in the waiting rooms and then did not show after the consultation) and refusals, application of questionnaires, and coding of forms] based on an instruction manual prepared by the core team of the project. Second, a pilot study was conducted on a random sample of 81 participants from three BHUs, intentionally selected among the 15 existing units, to test data collection procedures and to understand questions translated from other studies or adapted to the local context. Third, all pilot study participants were re-interviewed at an interval of seven to 10 days to analyze the temporal stability of the main variables of the study. The reproducibility of SB counseling was analyzed with percent agreement and Cohen’s kappa test, which showed 71% agreement and considerable Kappa value (0.72; *p* < 0.001). Fourth, data entry was performed by the field coordinator using the EpiData software (version 3.1, Odense, Denmark). Fifth, data verification and cleaning were performed using exploratory analysis in the SPSS software (version 26.0, SPSS Inc., Chicago, IL, USA) to identify possible typing errors and the presence of outliers and verify the distributions of all variables. Finally, each outlier variable was manually checked and corrected in the database.

### 2.8. Data Analysis 

The prevalence of counseling was described and analyzed between the categories of independent variables. The association was tested with Poisson regression to estimate the prevalence ratio (PR) and 95% Confidence Interval (95% CI) [38]. Initially, variance inflation factor (VIF) tests were applied and rejected the hypothesis of multicollinearity (1/VIF ≥ 0.81). After elaborating the bivariate analysis, multivariate analysis was performed following a hierarchical model [39]. Although there is no conceptual or analytical model of predictors for SB counseling, the hierarchical approach was applied to explore the data systematically [26]. Thus, all variables from the same or higher level with a *p*-value < 0.20 in the bivariate analysis were selected for adjustment in the multivariate model [22,26]. The final model was elaborated from the hierarchical structure with the following levels: level 1, sociodemographic characteristics; level 2, health conditions; level 3, access to health services; level 4, health risks behavior characteristics. Descriptive statistics (mean, standard deviation, 95% CI, median, interquartile range) were used to explore the SB in each activity. As the data were not normally distributed, the Mann-Whitney U test was used to compare the SB according to receiving SB counseling. It was decided to present the mean of hours to allow a possible comparison with other studies [7]. All analyses were performed using the STATA software and at a 5% significance level. The correction for design effect was performed using the command *svy* for accounting for the estimates of outcome variability. The sample power was analyzed a posteriori, and based on the predictor variables, the sample (n = 779) showed a power above 80% (β = 20%) and a 95% confidence level (α: 5%) to detect PR as significant [26,40].

## 3. Results

A total of 935 adults were approached, with a refusal rate of 14% (n = 134) and loss of 2% (n = 22). The final sample was 779 adults and was composed predominantly of women (69.8%), aged between 18–39 years (45.2%), not single (64.0%), white (73.0%), low economic level (71.2%), and BMI ≥ 25.0 kg/m^2^ (68.5%) (Table 1). 

Approximately 57% have ≥1 chronic condition, and 50% reported taking prescribed medications. Hypertension was the most prevalent disease (36%) (Figure 1). The prevalence of the BHU visits ≥3 times/year was 62.7%, and around 73% and 80% of participants reported performing between 0–9 min/wk of walking and MVPA, respectively. The highest SB quartile was ≥6.2 h/day, and the prevalence of SB counseling was 12.2% (95% CI: 10.1–14.7%) (Table 1). No significant difference was found in the prevalence of SB counseling between the BHU (*p* = 0.080) (Figure 2).

The mean of SB was 4.26 ± 3.47 h/day, highest in watching TV (1.88 ± 2.0 h/day) and lowest in studying (0.23 ± 0.87 h/day) (Table 2).

The SB watching TV (1.83 versus 2.25 h/day; *p* = 0.025), working (1.02 versus 2.05 h/day; *p* = 0.009), and daily (4.10 versus 5.48 h/day; *p* < 0.001) was higher among adults who received SB counseling (Table 3 and Figure 3).

In the bivariate analysis, female sex (PR: 1.62; 95% CI: 1.01–2.59), age group ≥60 yrs (PR: 1.62; 95% CI: 1.02–2.65), BMI ≥ 30.0 kg/m^2^ (PR: 2.89; 95% CI: 1.74–4.80), chronic diseases (PR ≥ 2.06; *p* < 0.05), visits to BHU ≥7 times/year (PR: 1.76; 95% CI: 1.10–2.82), walking in leisure time (PR ≥ 1.60; *p* < 0.05) and SB ≥ 6.2 h/day (4th quartile; PR ≥ 1.73, *p* < 0.05) were positively associated with counseling (Table 4).

In the multivariate analysis, SB counseling remained positively associated with the sociodemographic characteristics: female sex (PR: 1.77; 95% CI: 1.10–2.83) and age group ≥60 yrs (PR: 1.84; 95% CI: 1.14–2.98). The probability of receiving counseling was also higher among users who BMI ≥ 30.0 kg/m^2^ (PR: 2.60; 95% CI: 1.31–5.17) and who consumed ≥3 continuous medications (PR: 2.21; 95% CI: 1.06–4.59) and those sitting ≥1.8 h/day. A higher magnitude of association was found among those who remained seated ≥6.2 h/day (PR: 3.44; 95% CI: 1.88–6.31) (Table 4).

## 4. Discussion

This study aimed to identify the prevalence and examine the associated factors with SB counseling received from healthcare professionals in adults attending the PHC at BHU in Brazil. The prevalence of counseling was 12.2%, higher in women, older adults, and obese who consume ≥3 medications and remain in high SB. The quantitative approach and standardized methodology, with low selection bias, allowed us to explore the associations of sociodemographic characteristics, health conditions, access to health services, and risks behaviors with SB counseling in a representative sample of adults. Furthermore, this is the second study worldwide and the first in Latin America using this approach to identify the prevalence of counseling for reducing SB. Our study expands knowledge about the associated factors with SB counseling, especially in middle-income countries. From an international perspective, exploring these variables is essential so policymakers can develop policies and plan actions for counseling to reduce population-based SB in PHC.

Systematic reviews identified many studies that explored the association or the effect of counseling on increasing PA [16,17]. However, due to the high time spent on SB in Latin American adults [6,7], identifying the prevalence and associated factors with counseling to reduce SB in users of Health Systems is essential to monitor and encourage this practice by healthcare professionals in daily attendance. The prolonged SB is typical behavior and more prevalent than the recommended levels of PA in adults and should be the focus of PHC interventions [15,18,41].

This study evaluated a representative sample with an important sociodemographic characteristic, which could be, possibly, a more vulnerable population to SB in Latina American countries. Studies have shown a positive effect of counseling received to reduce SB [12,14,15,31]. Thus, our findings can support specific actions to improve PHC counseling in Brazil, specifically in the BHU [19]. However, SB counseling could be more specifically towards those Latin American population subgroups who spend more SB, such as men, the elderly, not single, those with lower education, low economic level, obese, and adults with chronic disease [6,7].

The low prevalence of SB counseling can be partly explained by possible forgetfulness among respondents regarding the received counseling, the structural characteristics of PHC, perception of professionals’ barriers to counseling, and the lack of protocols for counseling in clinical guidelines [42]. These reasons, among others, could result in different perceptions by users about what was advised. For example, users may not understand healthcare professionals’ guidance as an orientation but merely as an informal recommendation without a technical basis [16,29]. Also, professionals may have doubts or no knowledge about SB [42]. This also could be misunderstood by BHU users, confounding physical inactivity and SB. In addition, patients with acute diseases are not eligible for SB or PA counseling.

These results highlight the hypothesis that this public health orientation is underutilized and incipient in PHC [22,25]. Conversely, in Brazil, studies were conducted exclusively with BHU users and focused on PA counseling, of which the prevalence is 42% (±11%) [25]. The low prevalence of SB counseling could also be explained by the "recent" evidence about the deleterious effects of prolonged SB on public health (morbidity, mortality, costs) and what is independent of PA [8,9,10]. For example, many Brazilian physicians are recent graduates and possibly did not learn content related to the physiology of SB in their undergraduate [9,43]. Moreover, 39% are “general practitioners” (medical graduates only), and there are no “public health” specialists. Physicians with possible experience in public health or SB are specialized in “cardiology” (4.1%), “family and community medicine” (1.7%), “endocrinology and metabology” (1.4%), “geriatrics” (0.5%), “preventive and social medicine” (0.4%), “physical medicine and rehabilitation” and “sports medicine” (0.2% each). Only 20% of Brazilian physicians work at BHU or participate in a family health program or strategy team [43]. The PA Guidelines for the Brazilian Population, launched in 2021, can contribute to a better understanding and help healthcare professionals regarding counseling [41]. Moreover, professionals should encourage users to attend public open spaces in their leisure time and replace or reduce SB [15,18,41].

Women, older adults, obese, and who consume ≥3 medications are most likely to receive SB counseling. This finding can partly be explained by the fact that these populational subgroups make greater use of public health services [20]. These results are consistent with the literature, demonstrating the involvement of health services with this population, which requires further guidance due to the physiological changes affecting people of advanced age [15,27]. Another important fact related to this association is that healthcare professionals provide clinical advice for chronic diseases and depend on prescribed medication [25,27]. The association between medication use and SB counseling provides new insight into the interrelationship with pathologies and beliefs about the health status of PHC users. These associations could be analyzed in future studies on SB counseling and other predictors.

Our results showed that obesity was associated with counseling received. Shuval et al. [22] found similar results, and, according to the authors, the limited sample size might have inhibited the detection of a significant association between other variables and SB counseling [22]. However, the authors did not explain the possible causal relationship or biological plausibility between the variables. We hypothesized that the association could be partly explained because obesity is one of the main risk factors for chronic diseases, which may cause healthcare professionals to advise this population group more frequently [15,25]. Although it is beneficial to counsel obese individuals, it is essential that counseling is guaranteed to users regardless of diseases or any health condition, as PHC should be the main level of assistance for prevention, promotion, and health care actions [19]. However, it is essential to highlight that without nutritional counseling, healthy eating habits, and adequate and regular PA levels, SB counseling alone will not be effective for losing weight [17].

The highest association magnitude was found among those who spent SB ≥6.2 h/day, showing that the likelihood of receiving counseling was 3.4 times higher than those in the first quartile. This association can be explained by the fact that high SB daily is associated with greater risks of morbidity and mortality, which could cause healthcare professionals to advise this population subgroup more often, as adults spend a large part of their day sitting down [6,7,15], even if they practice regular PA [8,10]. These findings are positive and expected, as they may indicate that professionals are aware of the harmful effects on health due to high exposure to SB, for example, when spending long periods of the day sitting watching TV or working [13,15,41,44]. Partial results from our project showed that 58% of BHU users reported that health professionals advised PA based on their characteristics (i.e., sex, age, weight) [45].

Some limitations must be considered to interpret the results of this exploratory study. First, the sample was limited to adults attending a BHU from the urban area of a medium-sized metropolitan city in Southern Brazil. This city has a broad rural area (79% of the territory), comprising 12 BHU. Our results cannot be extrapolated to adolescents, those who have physical limitations for PA, or that live in other contexts (rural areas, small or large cities, states, or regions), and other countries in Latin America. Second, there are no PA professionals in PHC or PA programs at a BHU. This limitation and characteristics of BHUs and the structure of services offered can partially have altered the prevalence of SB counseling. Third, the recall bias inherent in the self-reported measure may have affected the prevalence of counseling or the time in SB [28]. Finally, the cross-sectional design limits the causality.

## 5. Conclusions

The prevalence of SB counseling was 12.2%, and populational groups more exposed to counseling were women, older adults, obese, who consumed medications, and those who spent long periods engaged in SB. The SB counseling is underutilized and incipient in PHC. Understanding these results can help managers and healthcare professionals implement and direct actions to reduce SB in adults through counseling.

As new insights from studies from other countries are added to the evidence, counseling may be most common for vulnerable populations with high SB and low PA levels. The PHC actions should be expanded, with healthcare professionals training to reduce SB and promote PA, emphasizing epidemiology and public health [23]. In addition, it is essential to develop counseling methods for the routine service that different healthcare professionals can use, with valid protocols to evaluate actions’ effectiveness. Future studies could explore the associated factors to SB in adults attending BHU and should assess the efficacy of different counseling actions regarding SB levels for different population groups due to differences in lifestyle between urban and rural areas, large and small cities, and countries in other income. Our results highlight that SB counseling in the PHC must be directed and identify sociodemographic characteristics, health conditions, access to health services, and risk behaviors [15,18].

## Figures and Tables

**Figure 1 ijerph-19-09963-f001:**
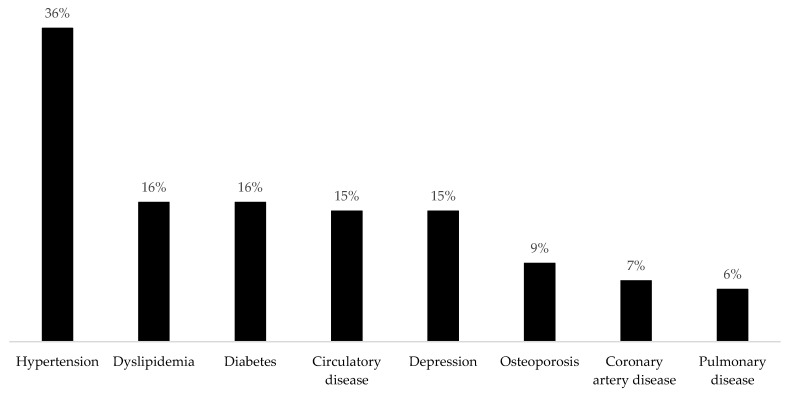
Prevalence of chronic disease of adults attending basic health units. São José dos Pinhais, Paraná, Southern Brazil, 2019 (n = 779).

**Figure 2 ijerph-19-09963-f002:**
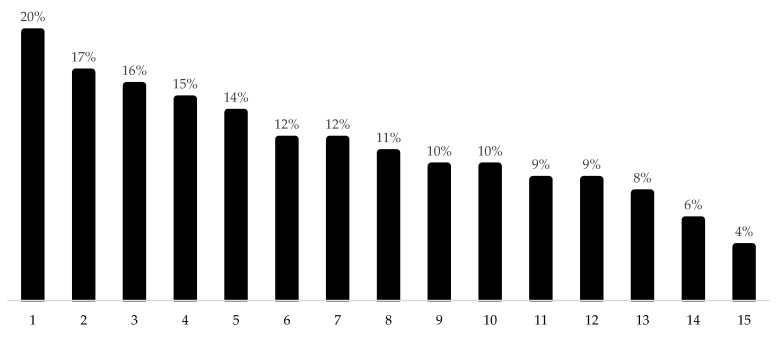
Prevalence of sedentary behavior counseling received from healthcare professionals by adults in each of the 15 basic health units evaluated. São José dos Pinhais, Paraná, Southern Brazil, 2019 (n = 779).

**Figure 3 ijerph-19-09963-f003:**
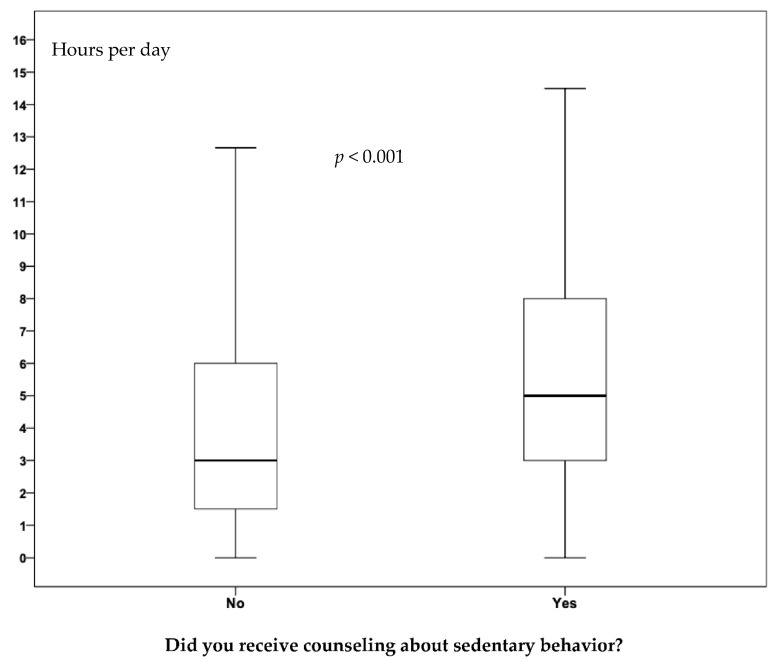
Median of sedentary behavior (total daily sitting time) according to counseling received from healthcare professionals by adults attending basic health units. São José dos Pinhais, Paraná, Southern Brazil, 2019 (n = 779).

**Table 1 ijerph-19-09963-t001:** Descriptive characteristics of adults attending basic health units. São José dos Pinhais, Paraná, Southern Brazil, 2019 (n = 779).

Variable	Categories	n	%	95% CI	Mean ± S. D.
Sociodemographic characteristics					
Sex	Male	235	30.2	27.0–33.5	–
	Female	544	69.8	66.5–73.0	
Age group (yrs)	18–39	346	45.2	41.7–48.7	43.7 ± 16.1
	40–59	283	36.9	33.6–40.4	
	≥60	137	17.9	15.3–20.8	
Marital status	Single	280	36.0	32.7–39.5	–
	Not single	497	64.0	60.5–67.3	
Skin color	White	566	73.0	69.8–76.0	–
	Non-white	209	27.0	24.0–30.2	
Economic level	Low	555	71.2	68.0–74.3	–
	High	224	28.8	25.7–32.0	
Health conditions					
Body Mass Index (kg/m^2^)	≤24.9	242	31.5	28.3–34.8	27.9 ± 5.2
	25.0–29.9	294	38.2	34.9–41.7	
	≥30.0	233	30.3	27.2–33.6	
Number of chronic diseases	0	334	42.9	39.4–46.4	1.2 ± 1.4
	1–2	311	39.9	36.5–43.4	
	≥3	134	17.2	14.7–20.0	
Number of prescribed medications	0	387	49.7	46.2–53.2	2.4 ± 2.6
	1–2	223	28.6	25.6–31.9	
	≥3	169	21.7	18.9–24.7	
Access to health services					
Frequency of visits to Basic Health Unit in the last year	0–2	291	37.4	34.0–40.8	5.5 ± 6.0
	3–6	299	38.4	35.0–41.8	
	≥7	189	24.3	21.4–27.4	
Health risks behavior					
Leisure-time physical activity					
Walking (min/wk)	0–9	567	72.8	69.6–75.8	50.2 ± 113.8
	10–149	108	13.9	11.6–16.5	
	≥150	104	13.4	11.1–15.9	
MVPA (min/wk)	0–9	626	80.4	77.4–83.0	15.2 ± 63.8
	10–149	55	7.1	5.5–9.1	
	≥150	98	12.5	10.4–15.1	
Sedentary behavior					
Sitting time (h/day)—quartiles *	0–1.7	193	24.8	21.9–27.9	4.3 ± 3.5
	1.8–3.3	192	24.6	21.7–27.8	
	3.4–6.1	197	25.3	22.4–28.5	
	≥6.2 (4th quartile)	197	25.3	22.4–28.5	
Did you receive counseling about sedentary behavior?	No	684	87.8	85.3–89.9	–
	Yes	95	12.2	10.1–14.7	

S. D.: standard deviation; * Sitting watching TV + working + commuting + using computer + studying.

**Table 2 ijerph-19-09963-t002:** Description of sedentary behavior of adults attending basic health units. São José dos Pinhais, Paraná, Southern Brazil, 2019 (n = 779).

Hours Per Day	Mean	S. D.	95% CI	Median	IQR
Watching TV	1.88	2.03	1.73–2.02	1.00	3.00
Working	1.14	2.44	0.97–1.32	0.04	1.00
Commuting	0.65	0.99	0.58–0.72	0.33	1.00
Using computer at home	0.37	1.05	0.30–0.44	0.00	0.00
Studying	0.23	0.87	0.18–0.29	0.00	0.00
Total	4.26	3.47	4.02–4.51	3.25	4.30

IQR: interquartile range, S. D.: standard deviation.

**Table 3 ijerph-19-09963-t003:** Sedentary behavior according to counseling received from healthcare professionals in adults attending Basic Health Units. São José dos Pinhais, Paraná, Southern Brazil, 2019 (n = 779).

	No (n = 684, 87.8%)					Yes (n = 95, 12.2%)					Sig.*
Hours Per Day	Mean	S. D.	95% CI	Median	IQR	Mean	S. D.	95% CI	Median	IQR	
Watching TV	1.83	2.02	1.67–1.98	1.00	3.00	2.25	2.07	1.82–2.68	2.00	2.20	**0.025**
Working	1.02	2.28	0.85–1.19	0.00	1.00	2.05	3.21	1.39–2.71	0.00	3.50	**0.009**
Commuting	0.64	0.96	0.57–0.71	0.33	1.00	0.71	1.17	0.47–0.95	0.33	1.00	0.532
Using computer at home	0.37	1.04	0.29–0.45	0.00	0.00	0.37	1.16	0.13–0.61	0.00	0.00	0.832
Studying	0.24	0.89	0.18–0.31	0.00	0.00	0.09	0.59	−0.03–0.21	0.00	0.00	0.118
Total	4.10	3.40	3.84–4.35	3.00	4.50	5.48	3.69	4.72–6.23	5.00	5.00	**<0.001**

IQR: interquartile range, S. D.: standard deviation, * Mann-Whitney U test.

**Table 4 ijerph-19-09963-t004:** Associated factors with sedentary behavior (SB) counseling received from healthcare professionals in adults attending basic health units. São José dos Pinhais, Paraná, Southern Brazil, 2019 (n = 779).

		Prevalence of SB Counseling (%)	Bivariate Analysis			Multivariate Analysis		
			PR	95% CI	*p* *	PR	95% CI	*p* *
Level 1—Sociodemographic characteristics								
Sex	Male	8.5	1			1 ^a^		
	Female	13.8	**1.62**	**1.01–2.59**	**0.044**	**1.77**	**1.10–2.83**	**0.017**
Age group (yrs)	18–39	10.3	1			1 ^b^		
	40–59	12.1	1.16	0.75–1.80	0.491	1.22	0.78–1.89	0.370
	≥60	17.0	**1.65**	**1.02–2.65**	**0.041**	**1.84**	**1.14–2.98**	**0.013**
Marital status	Single	11.1	1			–	–	–
	Not single	12.9	1.16	0.77–1.74	0.463	–	–	–
Skin color	White	12.2	1			–	–	–
	Non–white	12.4	1.02	0.70–1.56	0.925	–	–	–
Economic level	Low	12.1	1			–	–	–
	High	12.5	1.04	0.69–1.57	0.869	–	–	–
Level 2—Health conditions								
Body mass index (kg/m^2^)	<24.9 kg/m^2^	7.4	1			1 ^c^		
	25.0–29.9 kg/m^2^	8.8	1.19	0.67–2.12	0.556	1.30	0.62–2.73	0.482
	≥30.0 kg/m^2^	21.5	**2.89**	**1.74–4.80**	**<0.001**	**2.60**	**1.31–5.17**	**0.006**
Number of chronic diseases	0	7.5	1			1 ^d^		
	1–2	15.4	**2.06**	**1.30–3.26**	**0.002**	1.28	0.64–2.55	0.490
	≥3	16.4	**2.19**	**1.28–3.75**	**0.004**	0.86	0.38–1.93	0.706
Number of prescribed medications	0	7.2	1			1 ^e^		
	1–2	14.8	**2.05**	**1.27–3.29**	**0.003**	1.89	0.99–3.59	0.053
	≥3	20.1	**2.78**	**1.74–4.43**	**<0.001**	**2.21**	**1.06–4.59**	**0.035**
Level 3—Access to health services								
Frequency of visits to Basic Health Unit on the last year (times)	0–2	9.6	1			1 ^f^		
	3–6	11.7	1.22	0.76–1.95	0.414	0.98	0.61–1.55	0.916
	≥7	16.9	**1.76**	**1.10–2.82**	**0.019**	1.41	0.87–2.26	0.162
Level 4–Health risks behaviors								
Walking (min/wk)	0–9	10.4	1			1 ^g^		
	10–149	16.7	1.60	0.99–2.60	0.057	1.36	0.83–2.21	0.222
	≥150	17.3	**1.66**	**1.03–2.70**	**0.040**	1.43	0.90–2.29	0.132
MVPA (min/wk)	0–9	12.3	1					
	10–149	9.1	0.74	0.31–1.75	0.492	–	–	–
	≥150	13.3	1.08	0.62–1.87	0.787	–	–	–
Sedentary behavior (sitting time in h/day)—(quartiles)	0–1.7	6.2	1			1 ^h^		
	1.8–3.3	12.0	1.94	0.99–3.76	0.055	**2.03**	**1.03–4.01**	**0.041**
	3.4–6.1	10.7	1.73	0.87–3.39	0.121	**2.28**	**1.17–4.43**	**0.015**
	≥6.2 (4th quartile)	19.8	**3.19**	**1.72–5.89**	**<0.001**	**3.44**	**1.88–6.31**	**<0.001**

* Wald test; ^a^ adjusted by age group; ^b^ adjusted by sex; ^c^ adjusted by sex, age group, chronic diseas-es, and prescribed medications; ^d^ adjusted by sex, age group, BMI, and prescribed medications; ^e^ adjusted by sex, age group, BMI, and chronic diseases; ^f^ adjusted by sex, age group, BMI, chronic diseases, and prescribed medications; ^g^ adjusted by sex, age group, BMI, chronic diseases, pre-scribed medications, visits to basic health unit, and sedentary behavior; ^h^ adjusted by sex, age group, BMI, chronic diseases, prescribed medications, visits to basic health unit, and walking.

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
