# Peer review of "Sedentary Behavior Counseling Received from Healthcare Professionals: An Exploratory Analysis in Adults at Primary Health Care in Brazil"

_ijerph, 2022, doi:10.3390/ijerph19169963_

Round 1

Reviewer 1 Report

The study has limitations specifically with respect to work activities, it is particularly interesting to identify the reasons for the results with respect to working time and sedentary behavior.

The study could clarify this by specifying the type of work activities performed by the participants. Given the number of female participants, it would be interesting to determine if there is a bias related to the activities or occupation of the women, as it could be hypothesized that they are dedicated to household activities, given the number of hours declared as a base commitment, watching television.

Finally, it is necessary to establish in the context of the study the impact of work activities that imply sedentary behavior, or if there is a mix of urban and rural work activities, which would indicate greater physical activity.

Finally, it is important to be aware of the possible associations of factors such as drug use and pathologies, as this may have a bias of beliefs regarding individual health status. 

counseling as an activity of health systems implies a clear understanding of the nature of an individual's occupational and non-occupational activities in order to develop appropriate plans or strategies for managing sedentary behavior. 

Reviewer 2 Report

Comments to the authors

The authors conducted a study to investigate if sedentary behaviour (SB) counselling was delivered by healthcare professionals in primary healthcare in Brazil. They also investigated the characteristics of the patients who received (or not) the SB counselling. The topic is relevant, and the quality of the study is good. However, the current article has some drawbacks which should be fixed before publication.

Major comments

Introduction

Page 1, line 37. Please provide a definition of sedentary behavior.

Page 1, lines 39-40. The current article is about the Brazilian population. Therefore, the UH example is not relevant. It would be better to find a reference concerning the burden of SB in Brazil.

Page 2, lines 47-48. While SB and PA are different, the authors could also refer to the PA promotion within the healthcare system. Actually, the WHO recommends to promote PA within the health care system (action 3.2 of the Global action plan on physical activity 2018–2030: more active people for a healthier world. Geneva: World Health Organization; 2018.). In addition, SB (and PA) counselling may be not addressed at each healthcare contacts. Indeed, patients with acute diseases (e.g. fever) are not eligible for PA counselling. It must be acknowledged in the discussion.

Methods

Page 2, line 69. Why is the sample representative? Please explain.

Page 2, lines 74-75. The exclusion of the 12 rural BHU may lead to a bias. The patients (and their health issues) from the 12 rural BHU may be very different from the rest of the population.

Page 2, lines 79-84. Based on the present study design, I do not understand the need to calculate a sample size. Please explain the rationale for this sample size calculation. It would be interesting to know if the sample size calculation was made to have a representative sample. It would be interested if the characteristics of the local population (e.g. age, sex, occupation, diseases, etc.) were entered in the model to determine the sample size.  

Page 2, line 92. Please explain why the patients living in a rural area where excluded.

Page 2, line 93. The exclusion of patients with some physical limitation for PA practice could be disputed. The WHO published PA recommendations for those people. While it is not possible to include these patients, I recommend to add a commentary in the limitation section of the manuscript.

Page 3, line 94. Phonation limitation does not prevent comprehension of the questionnaire. Hearing impairment may prevent comprehension of the questionnaire. Phonation limitation may prevent to answer the questionnaire. It should be added here that the questionnaire was a face-to-face questionnaire.

Page 3, line 97. The authors should clarify the data collection. It is not clear if the interviewers read the questionnaires or if the participants read it.

Page 3, line 97. The interviews should have been done only before the consultation with the health professional to avoid a bias (health professional could do SBC because a study about this topic is running in the BHU where he or she is working). Please delete the interviews that were done after the consultation.

Page 3, line 101. Please indicate if the definition (with examples) of SB (and SBC) was provided to the participants. It would help them to identify if healthcare professionals provided SBC.

Page 4, line 149. How is it possible to record the losses in a cross-sectional study?

Page 4, line 156. Please comment if 70% of agreement is satisfying or not.

Page 4, line 171. Please consider to enter in the model only the variables with a p value <0.10.

Page 4, line 173. Please explain how were determined/ranked the levels.

Page 4, line 177. Please consider statistical adjustments for multiple comparisons.

Results

Page 4, line 185. Please explain the losses (i.e. they agreed to participate in the waiting rooms and then did not show after the consultation?).

Discussion

Page 11, line 253. Based on the presented data, it is not possible to determine if the sample is representative.

Page 11, lines 275-276. Aged physicians did not learn the effect of SB or insufficient PA. It is more likely that the recent graduates received a more updated training concerning SB and PA (https://pubmed.ncbi.nlm.nih.gov/35798540/).

Page 12, line 296. It is interesting to see that obese patients are more often counselled to change SB or PA. The authors should discuss that SB and/or PA without nutrition counselling will not help the patients to lose weight. Medical doctors may have been influenced (like the general population) by the false belief that obesity is driven (only) by SB or lack of PA. “Big Food corporations are spending billions of US$ on their strategy to claim that obesity is caused by physical inactivity. Their engagement with physical activity and public health organisations and professionals is part of their corporate social responsibility strategy. Their campaigns include techniques to evade regulation and to influence science, using methods similar to those used by tobacco corporations in the past” (https://www.thelancet.com/journals/lancet/article/PIIS0140-6736(14)60988-0/fulltext). Nevertheless, PA has however health benefits in obese patients (e.g. cardiovascular or psychological benefits).

Page 12, lines 307-315. It is highly probable that a large proportion of patients who spent SB>6.2 hours/day are obese and/or with other chronic diseases. I am wondering if the MDs are evaluating the SB and PA or if they counselled their patients based on their conditions. The authors should discuss if the MDs are evaluating the SB and PA.

Page 12, line 316. Please discuss the recall bias in the limitation section.

Page 12, line 321. This is not a limitation of the study. Lack of PA (not PE) professionals is a limitation of the BHU.

Page 12, line 332. The training of healthcare professionals is essential. The authors should more discuss it. They also should describe more the current SB counselling and the ideal SB counselling.

Minor comments

Abstract

Page 1, line 19. The first sentence of the abstract does not introduce well the topic. I suggest to change the sentence which should explain briefly the importance of the SBC in a healthcare perspective.

Page 1, lines 19-21. The sentence is difficult to follow. It could be as follow: « The study aimed to identify the prevalence of SB counselling provided to the patients by healthcare professionals at Primary Health Care in Brazil. It aimed also at identifying the characteristics of patients who received SB counselling.”

Page 1, line 21. Please use patients instead of adults when it is appropriate throughout the manuscript.

Page 2, line 56. Please briefly explain the content of the SB counselling.

Introduction

Page 1, lines 43-44. This sentence could be moved just after the first sentence of the introduction.

Page 2, line 53. Please consider to replace “and” by “which”.

Page 2, line 51. What are the basic health units? Please briefly explain.

Page 2, line 56. Please delete throughout the article the abbreviation SBC and replace it by SB counselling. Please avoid useless abbreviations.

Page 2, line 56. The prevalence of PA counselling is much more studied. Please find more details here: https://www.sport-sante.lu/wp-content/uploads/2022/02/DtschZSportmed_Originalia_Lion_Physical_Activity_Promotion_2022-1.pdf.

Page 2, lines 64-65. The sentence is difficult to follow. It could be as follow: « The study aimed to identify the prevalence of SB counselling provided to the patients by healthcare professionals at Primary Health Care in Brazil. It aimed also at identifying the characteristics of patients who received SB counselling.”

Methods

Page 3, line 110. Please explain the age cut off >60 years old? The WHO has different age categories (18-65; >65 years old). Please avoid useless abbreviation (e.g. yrs).

Page 3, line 112. Please consider to change “married” category by “not single”.

Page 3, line 112. I am not sure if “skin color” is appropriate wording. Please check.

Page 3, lines 122-123. Cancer is missing. Please explain.

Page 3, line 134. Please explain how the minutes/week of walking were investigated (part 1 questions 6 & 7 + part 2 questions 12 & 13 + part 4 questions 20 & 21: https://www.sralab.org/sites/default/files/2017-07/IPAQ_English_self-admin_long.pdf ?).

Page 3, line 137. Please explain the category 0-9 minutes/week. Two categories seem enough. I am not sure if the reference 33 is appropriate here.

Page 4, line 138. The IPAQ is also investigating the sitting time. Please explain why it was not used.

Page 4, line 165. “.(35)”. Please check the dot and add square brackets.

Page 4, line 169. Please consider: “…, the hierarchical approach was applied to explore … “

Results

Page 5, line 189. Please add the figure 1 in the table 1.

Page 5, line 193 Please add a figure illustrating the prevalence of SBC between the BHU.

Page 7, line 209. Please add the number of patients per category (no/yes).

Page 7, line 212. Please add the mean. Please a legend for the vertical axis. Please add the number of patients per category (no/yes).

Page 8, line 224. Please add “patients” after among.

Discussion

Page 11, lines 236-237. The sentence is difficult to follow. It could be as follow: « The study aimed to identify the prevalence of SB counselling provided to the patients by healthcare professionals at Primary Health Care in Brazil. It aimed also at identifying the characteristics of patients who received SB counselling.”

Page 11, line 246. Some additional information about PA counselling may be find here: https://academic.oup.com/heapro/article/34/4/877/5035043?login=false
